# Time-varying and tissue-dependent effects of adiposity on leptin levels: A Mendelian randomization study

**Tom G Richardson\*, Genevieve M Leyden, George Davey Smith**

MRC Integrative Epidemiology Unit (IEU), Population Health Sciences, Bristol Medical School, University of Bristol, Oakfield House, Oakfield Grove, Bristol, United Kingdom

## Abstract

**Background:** Findings from Mendelian randomization (MR) studies are conventionally interpreted as lifelong effects, which typically do not provide insight into the molecular mechanisms underlying the effect of an exposure on an outcome. In this study, we apply two recently developed MR approaches (known as 'lifecourse' and 'tissue-partitioned' MR) to investigate lifestage-specific effects and tissues of action in the relationship between adiposity and circulating leptin levels.

**Methods:** Genetic instruments for childhood and adult adiposity were incorporated into a multivariable MR (MVMR) framework to estimate lifestage-specific effects on leptin levels measured during early life (mean age: 10 y) in the Avon Longitudinal Study of Parents and Children and in adulthood (mean age: 55 y) using summary-level data from the deCODE Health study. This was followed by partitioning body mass index (BMI) instruments into those whose effects are putatively mediated by gene expression in either subcutaneous adipose or brain tissues, followed by using MVMR to simultaneously estimate their separate effects on childhood and adult leptin levels.

**Results:** There was strong evidence that childhood adiposity has a direct effect on leptin levels at age 10 y in the lifecourse ($\beta = 1.10$ SD change in leptin levels, 95% CI = 0.90–1.30, $p=6 \times 10^{-28}$), whereas evidence of an indirect effect was found on adulthood leptin along the causal pathway involving adulthood body size ($\beta = 0.74$, 95% CI = 0.62–0.86, $p=1 \times 10^{-33}$). Tissue-partitioned MR analyses provided evidence to suggest that BMI exerts its effect on leptin levels during both childhood and adulthood via brain tissue-mediated pathways ($\beta = 0.79$, 95% CI = 0.22–1.36, $p=6 \times 10^{-3}$ and $\beta = 0.51$, 95% CI = 0.32–0.69, $p=1 \times 10^{-7}$, respectively).

**Conclusions:** Our findings demonstrate the use of lifecourse MR to disentangle direct and indirect effects of early-life exposures on time-varying complex outcomes. Furthermore, by integrating tissue-specific data, we highlight the etiological importance of appetite regulation in the effect of adiposity on leptin levels.

**Funding:** This work was supported by the Integrative Epidemiology Unit, which receives funding from the UK Medical Research Council and the University of Bristol (MC_UU_00011/1).

\*For correspondence:
Tom.G.Richardson@bristol.ac.uk

## Editor's evaluation

The authors have recently developed two novel approaches to Mendelian randomization studies (1) lifecourse MR which relates the genetic instruments to the outcome, eg obesity, at different stages of life eg childhood, and adulthood and (2) tissue partitioned MR to determine if the genetic instruments have different effects on different tissues such as the brain and adipose tissue. They combine these approaches to investigate the influence of adiposity on circulating leptin in order to demonstrate the value/proof of concept of these techniques in extending the use of MR. This is very clearly

presented and well conducted work showing both important new methodology and clear-cut results and so providing compelling evidence on the impact of adiposity at age 10 and in middle life and the weight gain in between on leptin levels and that the effect is mediated via the brain.

## Introduction

People living with obesity have elevated levels of the peptide hormone leptin. This can be attributed to the amount of leptin in circulation being proportional to the amount of adipose tissue that an individual has (*Obradovic et al., 2021*). After being secreted by fat cells in adipose tissue, leptin predominantly acts in the hypothalamus as a major regulator of energy balance (*Morrison, 2009*). Likewise, both neural and adipose tissues are known to play an important role in the molecular etiology of body mass index (BMI), which is conventionally used to clinically diagnose obesity. However, despite being a cost-effective approach to routinely measure adiposity at scale, BMI is a construct that captures multiple heterogeneous subcomponents. This is epitomized by previous investigations into the functional genes that exert their effects on adiposity via their expression in brain and adipose tissues (*Timshel et al., 2020*; *Rask-Andersen et al., 2019*), highlighting the divergent pathways that exist between BMI and downstream complex traits.

Mendelian randomization (MR) is a causal inference technique that can exploit the random segregation of genetic variants within a population to evaluate the genetically predicted effects of modifiable exposures on complex outcomes (*Davey Smith and Ebrahim, 2003*; *Richmond and Davey Smith, 2022*). We recently extended the principles of MR to evaluate the separate effects of molecular subcomponents of BMI using genetic variants partitioned by their impact on gene expression derived from subcutaneous adipose tissue (SAT) and brain tissue (known as 'tissue-partitioned MR') (*Leyden et al., 2022*). Using this approach suggested that the brain-derived variants are predominantly responsible for driving the effect of BMI on cardiometabolic disease outcomes (*Leyden et al., 2022*), which we postulate is due to their involvement in appetite regulation and energy expenditure. These in turn influence the timelines and sites of adipose tissue distribution, which have more adverse consequences at the same level of BMI as do the processes influenced by the SAT-derived variants. Conversely, the SAT-derived set of instruments were predominantly responsible for the effect of BMI on outcomes such as endometrial cancer (*Leyden et al., 2023*). Differences in components of adiposity at the same level of BMI have previously been demonstrated with these instruments (*Leyden et al., 2022*), preserving the gene–environment equivalence assumption in MR (*Davey Smith, 2012*).

In this study, we sought to dissect the causal pathway between BMI and circulating leptin using tissue-partitioned MR by leveraging these brain- and adipose tissue-derived sets of instruments. However, investigating this hypothesis is made even more challenging given that associations between BMI and leptin levels have been reported as early in life as childhood (*Shalitin and Phillip, 2003*). Therefore, to further develop insight into the relationship between BMI and circulating leptin, we additionally applied another extension that we have developed in recent years known as 'lifecourse MR' (*Richardson et al., 2020*). This approach allows the independent effects of childhood and adult adiposity to be simultaneously estimated on an outcome that can also be measured at separate timepoints in the lifecourse (*Richardson et al., 2022b*; *Sanderson et al., 2022*), such as childhood (mean age: 10 y) and adulthood (mean age: 55 y) measures of circulating leptin levels in this study. Taken together, we aimed to conduct a proof-of-concept study for these novel extensions of the conventional MR approach.

## Methods
### Lifecourse Mendelian randomization

Lifecourse MR has been described in detail previously (*Richardson et al., 2020*). Briefly, genetic instruments derived in the UK Biobank study based on self-reported body size at age 10 and measured adulthood BMI (*Richardson et al., 2020*) have been shown to separate clinically measured childhood and adult BMI in three independent cohorts (Avon Longitudinal Study of Parents and Children [ALSPAC] [*Richardson et al., 2020*], the Young Finns Study [*Richardson et al., 2021*], and the Trøndelag Health (HUNT) study [*Brandkvist et al., 2021*]). Both univariable and multivariable MR analyses on childhood leptin were conducted in a one-sample setting using individual-level data from ALSPAC

by analyzing genetic risk scores with adjustment for age and sex. Univariable MR analyses to estimate total effects on adulthood leptin were undertaken in a two-sample setting using the inverse variance weighted (IVW) method (*Burgess et al., 2013*), as well as the weighted median and MR-Egger methods (*Bowden et al., 2015*; *Bowden et al., 2016*). Multivariable MR analyses on adulthood leptin were also performed in a two-sample setting to estimate the direct and indirect effects of childhood body size (*Sanderson et al., 2019*).

## Tissue-partitioned Mendelian randomization

Instrument derivation and methodology for tissue-partitioned MR has also been described in detail previously (*Leyden et al., 2022*). In brief, 86 independent adult BMI-associated instruments provided evidence of sharing a causal variant with proximal gene expression in subcutaneous adipose tissue through genetic colocalization analyses (based on a posterior probability [PPA] > 0.8). The same approach provided evidence that the effects of 140 adult BMI instruments are putatively mediated by a nearby gene's expression in brain tissue. These two sets of genetic variants have near identical average effect sizes on BMI (adipose = 0.0148 and brain = 0.0149 standard deviation [SD] change in BMI per effect allele), although the manner in which they relate to other anthropometric traits can markedly differ. For instance, we previously found that the brain tissue instruments were more strongly correlated with waist-to-hip ratio compared to the adipose–tissue instruments ($r = 0.733$ and $r = 0.445$, respectively, $p_{comparison}=0.001$). Likewise, the brain tissue set was more highly correlated than the adipose set with a measure of visceral adipose tissue from the UK Biobank study ($r = 0.554$ and $r = 0.254$, respectively, $p_{comparison}=0.009$), further suggesting that the way in which these tissue-partitioned sets of variants exert their effects on BMI likely varies in terms of biological pathways.

Univariable and multivariable MR analyses on childhood leptin measured in ALSPAC were undertaken as above in a one-sample setting by aggregating the tissue-partitioned BMI instruments as genetic risk scores and analyzing childhood leptin with adjustment for age and sex. Estimates on adulthood leptin were initially evaluated in a two-sample setting using the three univariable MR methods mentioned above, and subsequently using an extension of multivariable MR with instruments weighted by their PPA values for each tissue type (*Leyden et al., 2023*). An overview of datasets used for exposures and outcomes analyzed in this study can be found in *Supplementary file 1a*. All analyses were conducted using the 'TwoSampleMR' R package (*Hemani et al., 2018*).

## Childhood measures of circulating leptin and fat distribution

The ALSPAC is a population-based cohort investigating genetic and environmental factors that affect the health and development of children. The study methods are described in detail elsewhere (*Boyd et al., 2013*; *Fraser et al., 2013*). In brief, 14,541 pregnant women residents in the former region of Avon, UK, with an expected delivery date between April 1, 1991, and December 31, 1992, were eligible to take part in ALSPAC. Of these initial pregnancies, there was a total of 14,676 fetuses, resulting in 14,062 live births and 13,988 children who were alive at 1 y of age. Please note that the study website contains details of all the data that is available through a fully searchable data dictionary and variable search tool and reference the following webpage: http://www.bristol.ac.uk/alspac/researchers/our-data/. Written informed consent was obtained for all study participants. Ethical approval for this study was obtained from the ALSPAC Ethics and Law Committee and the Local Research Ethics Committees. Childhood measures of circulating leptin levels were obtained from non-fasting blood samples taken from ALSPAC participants at mean age 9.9 y (range = 8.9–11.5 y). Leptin was measured by an in-house ELISA validated against commercial methods. Analyses in ALSPAC were undertaken on a final sample size of 4155 individuals after removing those without genetic data and withdrawn consent.

A further sensitivity analysis of body composition was conducted on dual-energy X-ray absorptiometry (DXA) and skinfold measures obtained from the ALSPAC study. Trunk and total body DXA measures of lean, bone, and fat mass were obtained for ALSPAC participants at mean age 9.9 y (range = 8.9–11.5 y) as above. Skinfold thickness measures were available for 576 ALSPAC participants assessed at the 61-mth clinic (mean age 5.2 y) by skinfold caliper. The mean of three measurements was obtained for biceps, triceps, subscapular and suprailiac skinfold thickness.

## Adulthood estimates for circulating leptin and fat distribution

Genetic estimates on adulthood circulating leptin levels were obtained from a study of 10,708 individuals enrolled in the Fenland study. Full details have been described elsewhere (*Pietzner et al., 2021a*). Leptin was measured using the SomaScan v4 assay, which applies single-stranded oligonucleotides aptamers with specific binding affinities to protein targets. This assay has also been reported to provide a very highly correlated measure of circulating leptin using the antibody-based Olink platform (*r* = 0.95) (*Pietzner et al., 2021b*). The Fenland study was approved by the National Health Service (NHS) Health Research Authority Research Ethics Committee (NRES Committee-East of England Cambridge Central, ref 04/Q018/19). All participants provided written informed consent. Although effect estimates from these summary-level data can be interpreted as a 1-SD change in normalized plasma leptin levels, we note that direct comparisons between estimates derived from childhood measures of leptin in ALSPAC should not be drawn given the various sources of heterogeneity between these measures.

We also obtained genetic estimates on adulthood circulating leptin levels from a previously conducted large-scale study of 35,559 individuals enrolled in the deCODE Health study (mean age: 55 y, SD: 17 y). Full details on that study have been reported previously (*Ferkingstad et al., 2021*). Briefly, circulating leptin was measured from plasma samples using the SomaScan version 4 assay by SomaLogic. All participants from this study who donated samples gave informed consent, and the National Bioethics Committee of Iceland approved the study, which was conducted in agreement with conditions issued by the Data Protection Authority of Iceland (VSN_14-015).

Genome-wide effect estimates on measures of fat distribution were obtained from a recent study on 38,965 UK Biobank participants who analyzed MRI-derived measures of visceral, abdominal subcutaneous, and gluteofemoral fat tissue volumes (*Agrawal et al., 2022*).

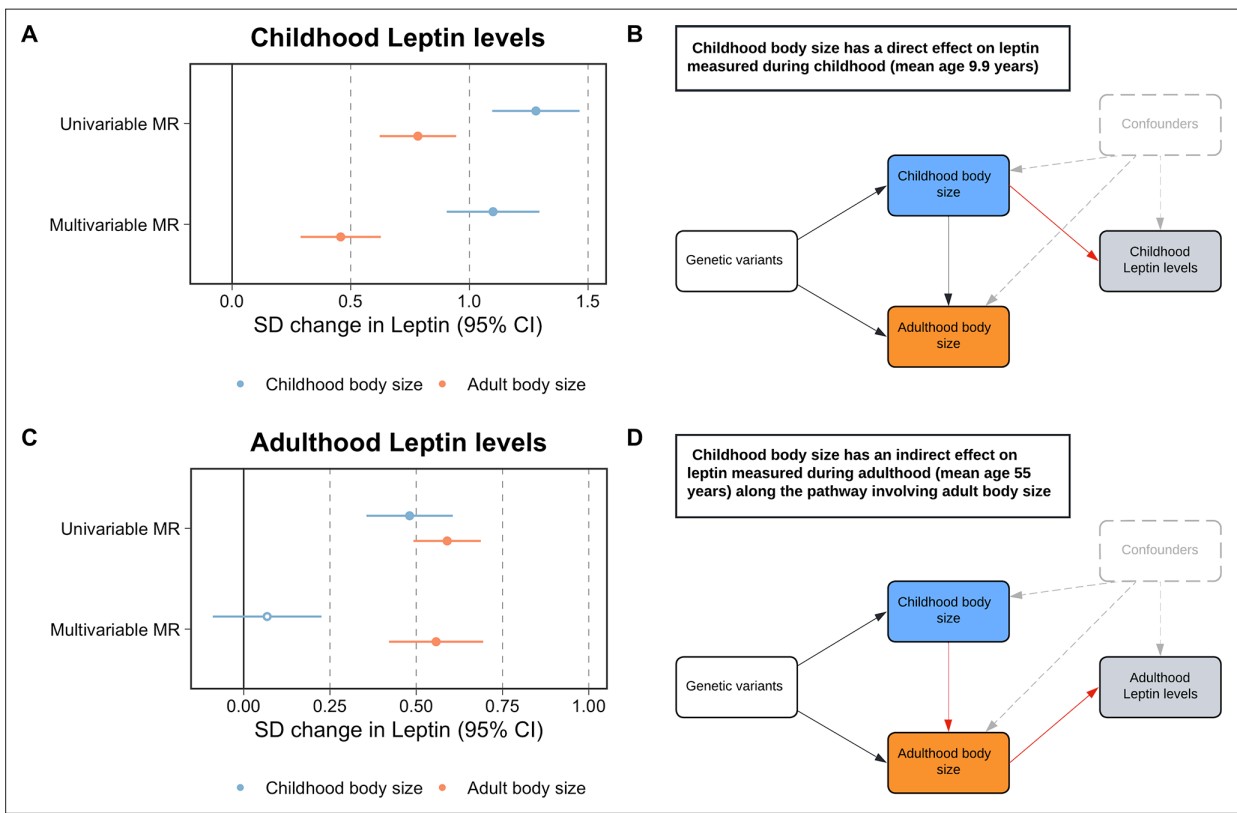

**Figure 1.** Forest plots and schematic diagrams for Lifecourse Mendelian randomization analysis. (**A**) A forest plot illustrating the direct effect of childhood body size on circulating leptin levels measured during childhood (mean age: 9.9 y) using individual-level data from the Avon Longitudinal Study of Parents and Children (ALSPAC). (**B**) provides the corresponding schematic diagram for ALSPAC results. (**C**) A forest plot depicting the indirect effect of childhood body size on adulthood measured leptin levels using data from the deCODE Health study (mean age: 55 y) as portrayed in the schematic diagram presented in panel (**D**). 'Genetic variants' refers to instruments for both exposures in the model. The red arrow indicates the causal pathway being assessed. MR, Mendelian randomization. Data underlying this figure can be found in *Supplementary file 1b and d*.

## Results

### Disentangling direct and indirect effects of childhood adiposity on early and mid-life measures of leptin levels

We firstly conducted univariable MR to estimate the total effect of childhood adiposity on circulating leptin using data measured in ALSPAC participants at mean age 9.9 y in the lifecourse (β = 1.28 SD change in leptin levels per change in body size category, 95% CI = 1.10–1.46, p=2 × 10⁻⁴¹). This was followed by applying multivariable MR, which provided evidence that childhood body size has a direct effect on increased leptin levels at this point in the lifecourse (β = 1.10, 95% CI = 0.90–1.30, p=6 × 10⁻²⁸) (*Figure 1A and B*, *Supplementary file 1b*). Both childhood and adult adiposity likewise provided evidence of a total effect on circulating leptin measured in adulthood (β = 0.48, 95% CI = 0.36–0.61, p=5 × 10⁻¹⁴ and β = 0.59, 95% CI = 0.49–0,69, p=2 × 10⁻³², respectively) (*Supplementary file 1c*). However, multivariable MR analyses suggested that childhood adiposity indirectly influences leptin levels measured during adulthood along the causal pathway involving adult body size (β = 0.56, 95% CI = 0.42–0.69, p=1 × 10⁻¹⁵) (*Figure 1C and D*, *Supplementary file 1d*). Similar patterns were found analyzing circulating leptin levels using data from the Icelandic population (*Supplementary file 1e*).

A further analysis of childhood body composition was conducted to investigate the residual positive effect of adult adiposity on leptin measured in childhood found in the previous multivariable MR analysis. Similarly, this analysis provided evidence that the adult effect attenuated with respect to the childhood adiposity score when evaluating measures of fat mass obtained by DXA scan (e.g. trunk fat; childhood body size: β = 1.42, 95% CI = 1.22–1.61, p=2.69 × 10⁻⁴⁷ and adult body size: β = 0.60, 95% CI = 0.44–0.77, p=1.6 × 10⁻¹²) (*Supplementary file 1f*). Additionally, an evaluation of skinfold measures provided further evidence of a direct effect of childhood adiposity on fat mass (e.g. childhood: β = 1.00, 95% CI = 0.40–1.60, p=0.001 and adult: β = −0.12, 95% CI = −0.65–0.42, p=0.67) (*Supplementary file 1g*).

### Separating the tissue-partitioned effects of body mass index on leptin levels measured during childhood and adulthood

Univariable MR analyses provided evidence of a total effect of adiposity on leptin levels measured during childhood based on analyses using the adipose tissue (β = 0.61, 95% CI = 0.13–1.08, p=0.01) and brain tissue (β = 0.86, 95% CI = 0.40–1.31, p=2 × 10⁻⁴) partitioned instruments. In a multivariable setting, the brain tissue-derived component of BMI predominated in the model (β = 0.79, 95% CI = 0.22–1.36, p=6 × 10⁻³), whereas the adipose tissue-derived estimate attenuated to include the null (β = 0.12, 95% CI = −0.48–0.71, p=0.70) (*Figure 2A and B*, *Supplementary file 1h*). Analyses on adulthood measured leptin also provided strong evidence of a total effect based on adipose-and brain tissue-derived estimates (β = 0.39, 95% CI = 0.21–0.57, p=3 × 10⁻⁵ and β = 0.42, 95% CI = 0.28–0.56, p=2 × 10⁻⁹, respectively). Similar to findings for childhood leptin, subcutaneous adipose tissue-derived estimates attenuated substantially more (β = 0.14, 95% CI = −0.14–0.42, p=0.33) compared to the estimates derived using the brain tissue instrument set (β = 0.38, 95% CI = 0.14–0.62, p=2 × 10⁻³) in a multivariable setting (*Figure 2C and D*, *Supplementary file 1i and j*). Similar conclusions were drawn based on data from the Icelandic population (*Supplementary file 1k*). Cochran's Q-statistics for these analyses, as well as those derived in lifecourse MR analyses, can be found in *Supplementary file 1l*, along with intercept terms for MR-Egger analyses, which did not suggest horizontal pleiotropy may be biasing conclusions.

As a further sensitivity analysis to characterize the causal pathway between these tissue-partitioned variants and fat distribution, we found that in particular the brain-expressed instruments have a predominating effect on visceral fat volume (β = 0.51, 95% CI = 0.30–0.72, p=2 × 10⁻⁶) compared to the subcutaneous adipose instruments (β = 0.07, 95% CI = −0.18–0.32, p=0.57) (*Figure 2—figure supplement 1*, *Supplementary file 1m*).

## Discussion

In this study, we applied two recent approaches in the MR paradigm, known as lifecourse and tissue partitioned MR, to investigate the influence of adiposity on circulating leptin levels as an exemplar

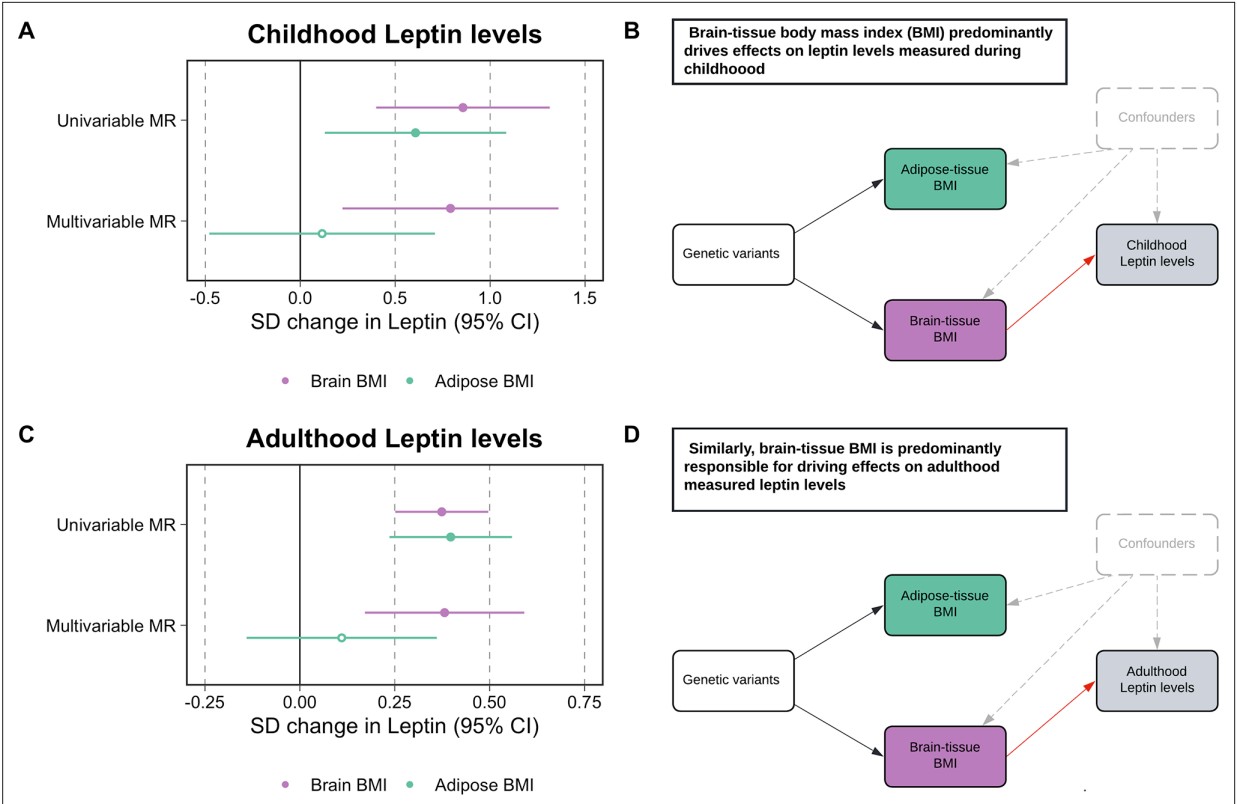

**Figure 2.** Forest plots and schematic diagrams for Tissue-partitioned Mendelian randomization analysis. (**A**) A forest plot illustrating the attenuation of the adipose tissue-instrumented body mass index (BMI) effect on childhood leptin levels, whereas estimates for brain tissue-instrumented BMI remained robust. (**B**) The corresponding schematic diagram for this finding. (**C**) A forest plot displaying similar conclusions to the estimates found in childhood, but derived using adulthood measured leptin levels. (**D**) provides the corresponding schematic diagram for estimates portrayed in panel (**C**). 'Genetic variants' refers to instruments for both exposures in the model. The red arrow indicates the causal pathway being assessed. MR, Mendelian randomization. Data underlying this figure can be found in *Supplementary file 1h and j*.

The online version of this article includes the following figure supplement(s) for figure 2:

**Figure supplement 1.** Forest plots depicting the univariable and multivariable Mendelian randomization (MR) effect estimates of adipose tissue-instrumented body mass index (BMI) (yellow) and brain tissue-instrumented BMI (red) on adulthood measures of (**A**) abdominal subcutaneous fat volume, (**B**) visceral adipose fat volume, and (**C**) gluteofemoral fat volume.

relationship to demonstrate the value of these techniques. Whilst there is irrefutable evidence that having a higher BMI increases circulating levels of leptin, these two extensions of MR methodology provide insight into lifecourse-specific effects and the tissues of action underlying this etiological relationship. We propose that these approaches can be conducted to build upon the findings by conventional MR analyses, which use naturally occurring genetic variants as causal anchors to help establish and estimate the causal effect of modifiable exposures on complex outcomes and disease endpoints.

Our application of lifecourse MR in this work supports a breadth of previous research using this technique, which highlights the importance of taking into consideration the age at which data from participants for exposure and outcome datasets are analyzed. Findings from this study provide evidence that adiposity in childhood exerts a direct effect on leptin levels when measured in early life, corroborating findings from observational studies (*Shalitin and Phillip, 2003*). Conversely, evidence of an indirect effect along the causal pathway involving adulthood adiposity was found when analyzing leptin measured in later life. This provides a powerful proof of concept for this approach, which has previously found evidence of a direct effect of increased childhood adiposity on lower risk of breast cancer (*Richardson et al., 2020*), elevated risk of type 1 diabetes (*Richardson et al., 2022a*), and an influence on cardiac structure (*O'Nunain et al., 2022*). Evidence supporting indirect effects using this approach, as in this study for adulthood leptin, has been found previously for various cardiovascular

(*Power et al., 2021*) and site-specific cancer endpoints (*Mariosa et al., 2022*; *Papadimitriou et al., 2023*).

Additionally, in the multivariable MR analysis of childhood adiposity on childhood leptin, we found that the effect of adulthood adiposity, despite attenuating in comparison to univariable MR estimates, still provided some evidence of an indirect influence. However, within a lifecourse model an effect of adulthood adiposity on childhood leptin levels is impossible given that an adulthood risk factor cannot influence a trait measured at an earlier stage in the lifecourse. We therefore investigated this residual effect of the adulthood adiposity genetic score on in-depth measures of childhood adiposity using data from the age 10 clinic of the ALSPAC study. Similar evidence of a residual effect of the adulthood genetic score on childhood measured DXA-assessed fat mass was found in these analyses, whereas a direct effect of childhood adiposity was found on skinfold measures. These analyses further complement the study design of leveraging both individual- and summary-level data within an MR framework to develop more granular insight into the underlying etiological pathways between risk factors and complex traits. This residual influence of the adulthood BMI score on childhood leptin suggests that the aspects of childhood adiposity that relate to the adulthood genetic score in the multivariable analyses independently of the childhood score importantly influence leptin in childhood. In similar analyses for childhood vitamin D no such relationship was seen (*Richardson et al., 2022b*). This suggests that different aspects of childhood body composition influence circulating vitamin D and leptin levels, and demonstrates how cognisance of the gene-environment equivalence assumption in MR can motivate improved understanding of genotype to phenotype associations.

Tissue-partitioned MR analyses in this work suggest that the subcomponent of BMI proxied using genetic variants whose effects are putatively mediated via gene expression in brain tissue are predominantly responsible for driving effects of adiposity on leptin during both childhood and adulthood. We postulate that these findings highlight a role for appetite regulation and energy expenditure mechanisms as of fundamental importance in the effect of adiposity on leptin levels. This corroborates findings from the literature, which suggest that being overweight has a downstream consequence on elevated leptin levels due to leptin resistance occurring in the central nervous system, particularly the hypothalamus (*Gruzdeva et al., 2019*). This is influenced by factors such as blood–brain barrier permeability, which results in leptin failing to suppress appetite and consequently leads to circulating hyperleptinemia amongst patients with obesity (*Izquierdo et al., 2019*). Taken together with the evidence of an indirect effect highlighted by our lifecourse MR analysis, these findings suggest that leptin may have long-term consequences for appetite suppression. This in turn has an influence on excess body weight, which may start in childhood and then can be sustained into later life. Future research in this space should investigate large-scale genome-wide association studies (GWAS) outcomes related to subcutaneous adipose tissues (e.g. skinfold thickness), particularly given that this is where leptin is primarily produced (*Russell et al., 1998*).

Fractionating genetic instruments for the same exposure using molecular datasets has important considerations for the (often overlooked) gene–environment equivalence assumption in MR (*Davey Smith, 2012*). This states that the effect of germline genetic perturbations should have the same downstream consequence on outcomes as if they were caused by the modifiable exposures themselves. For the adipose- and brain tissue-partitioned sets of instruments used in this study, we found previously that the manner in which they relate to downstream disease and complex traits can drastically differ despite having almost identical average effect estimates on BMI as an exposure. In particular, this finding underlines the heterogeneous nature of BMI as a lifestyle risk factor and highlights that this human-derived construct likely captures various causal pathways underlying the relationship between anthropometry and complex outcomes. For example, we previously found evidence that the brain tissue-derived variants are predominantly responsible for the effect of BMI on both cigarette smoking and lung cancer (*Leyden et al., 2023*). We emphasize that future application of tissue-partitioned MR should carefully consider both the exposure and functionally relevant tissue types being investigated, as well as ensuring that the derived instrument sets are equally predictive of the exposure (*Leyden et al., 2023*). In particular, applications of a combined approach using both lifecourse and tissue-partitioned MR will likely be most powerful for exposures that have been analyzed by GWAS in large samples at distinct timepoints in the lifecourse (e.g. childhood and adulthood) where the functionally important tissue types have been well characterized by previous studies.

We note that both lifecourse and tissue-partitioned MR have important caveats. For instance, the childhood and adult body size instruments used to disentangle direct and indirect effects in this study do not provide insight into other timepoints over the lifecourse (e.g. adolescence). Future efforts should focus on deriving instruments to separate effects of age-specific adiposity at more granular windows over the lifecourse. Moreover, the childhood adiposity instruments are based on recall data, which is why they required validation in three independent cohorts as described in the 'Methods' section. In addition, tissue-partitioned instruments were derived using bulk tissue in this work due to the availability of data and therefore do not take into account cell-type heterogeneity (*Glastonbury et al., 2019*; *Prince et al., 2021*). Furthermore, as with all estimates derived from MR, triangulating findings from other orthogonal lines of evidence derived using different study design and datasets provides the most robust conclusions for the approaches applied in this work (*Munafò and Davey Smith, 2018*).

In summary, our findings highlight a putative role for genes expressed in neural tissues in the etiology of adiposity and leptin levels during both childhood and adulthood. Furthermore, this innovative study provides a proof of concept into how the principles of MR can be adapted to investigate the hypotheses outside the scope of how this causal inference technique was originally conceived (*Davey Smith and Ebrahim, 2003*).

## Acknowledgements

We are extremely grateful to all the families who took part in this study, the midwives for their help in recruiting them, and the whole ALSPAC team, which includes interviewers, computer and laboratory technicians, clerical workers, research scientists, volunteers, managers, receptionists, and nurses. The UK Medical Research Council and Wellcome (grant ref: 217065/Z/19/Z) and the University of Bristol provide core support for ALSPAC. Consent for biological samples has been collected in accordance with the Human Tissue Act (2004). GWAS data was generated by Sample Logistics and Genotyping Facilities at Wellcome Sanger Institute and LabCorp (Laboratory Corporation of America) using support from 23andMe. This research was conducted at the NIHR Biomedical Research Centre at the University Hospitals Bristol NHS Foundation Trust and the University of Bristol. The views expressed in this publication are those of the author(s) and not necessarily those of the NHS, the National Institute for Health Research or the Department of Health. This publication is the work of the authors and TGR will serve as guarantor for the contents of this paper. This work was supported by the Integrative Epidemiology Unit, which receives funding from the UK Medical Research Council and the University of Bristol (MC_UU_00011/1).

## Additional information

### Competing interests

Tom G Richardson: TGR is an employee of GlaxoSmithKline outside of this work. The other authors declare that no competing interests exist.

### Funding

| Funder | Grant reference number | Author |
|--------|------------------------|--------|
| Medical Research Council | MC_UU_00011/1 | George Davey Smith |

The funders had no role in study design, data collection and interpretation, or the decision to submit the work for publication.

### Author contributions

Tom G Richardson, Data curation, Formal analysis, Visualization, Methodology, Writing - original draft, Writing - review and editing; Genevieve M Leyden, Data curation, Formal analysis, Methodology, Writing - review and editing; George Davey Smith, Conceptualization, Supervision, Funding acquisition, Methodology, Writing - review and editing

**Author ORCIDs**
Tom G Richardson http://orcid.org/0000-0002-7918-2040
George Davey Smith http://orcid.org/0000-0002-1407-8314

**Ethics**
ALSPAC: Written informed consent was obtained for all study participants. Ethical approval for this study was obtained from the ALSPAC Ethics and Law Committee and the Local Research Ethics Committees. DeCODE: All participants from this study who donated samples gave informed consent, and the National Bioethics Committee of Iceland approved the study, which was conducted in agreement with conditions issued by the Data Protection Authority of Iceland (VSN_14-015).

**Decision letter and Author response**
Decision letter https://doi.org/10.7554/eLife.84646.sa1
Author response https://doi.org/10.7554/eLife.84646.sa2

## Additional files

### Supplementary files

• Supplementary file 1. Supplementary tables. (**a**) Exposures and outcomes in this study. (**b**) Multivariable Mendelian randomization using lifecourse instruments on childhood leptin. (**c**) Univariable Mendelian randomization using lifecourse instruments. (**d**) Multivariable Mendelian randomization using lifecourse instruments on adult leptin. (**e**) Univariable and multivariable lifecourse Mendelian randomization analyses using adulthood leptin data derived from deCODE. (**f**) Multivariable Mendelian randomization analysis of DXA-derived measures in ALSPAC. (**g**) Multivariable Mendelian randomization analysis of skinfold measures in ALSPAC. (**h**) Multivariable Mendelian randomization using tissue-partitioned instruments on childhood leptin. (**i**) Univariable Mendelian randomization using tissue-partitioned instruments. (**j**) Multivariable Mendelian randomization using tissue-partitioned instruments on adult leptin. (**k**) Univariable and multivariable tissue-partitioned Mendelian randomization analyses using adulthood leptin data derived from deCODE. (**l**) Cochran's Q-statistics and MR-Egger intercept terms. (**m**) Multivariable Mendelian randomization using tissue-partitioned instruments on adult measures of fat distribution.

• MDAR checklist

### Data availability

All individual level data analysed in this study was obtained from the ALSPAC study which is not allowed to be deposited in a public repository. However, all data can be accessed via an application to ALSPAC which requires approval from executive committee (http://www.bristol.ac.uk/alspac/researchers/access/). Summary-level data on adulthood leptin levels were provided by the deCODE Health study which can be found at (https://download.decode.is/form/folder/proteomics). All other data analysed in this study is based on summary-level results as referenced throughout.

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
