## [Editor Report]

The authors have recently developed two novel approaches to Mendelian randomization studies (1) lifecourse MR which relates the genetic instruments to the outcome, eg obesity, at different stages of life eg childhood, and adulthood and (2) tissue partitioned MR to determine if the genetic instruments have different effects on different tissues such as the brain and adipose tissue. They combine these approaches to investigate the influence of adiposity on circulating leptin in order to demonstrate the value/proof of concept of these techniques in extending the use of MR. This is very clearly presented and well conducted work showing both important new methodology and clear-cut results and so providing compelling evidence on the impact of adiposity at age 10 and in middle life and the weight gain in between on leptin levels and that the effect is mediated via the brain.

---

## [Decision Letter]

**Decision letter after peer review:**

Thank you for submitting your article "Time-varying and tissue-dependent effects of adiposity on leptin levels: a Mendelian randomization study" for consideration by *eLife*. Your article has been reviewed by 2 peer reviewers, including Edward D Janus as Reviewing Editor and Reviewer #1, and the evaluation has been overseen by Matthias Barton as the Senior Editor. The following individual involved in the review of your submission has agreed to reveal their identity: Despoina Manousaki (Reviewer #2).

The reviewers have discussed their reviews with one another, and the Reviewing Editor has drafted this letter to help you prepare a revised submission.

Essential revisions:

1) It might not fit directly with the proof of concept aim of the manuscript, but this study could be a nice opportunity to test the opposite association, ie if leptin affects adiposity at different life stages. The relationship between BMI and leptin levels is complex as nicely depicted in lines 248-253 of the discussion. In the introduction, it is stated that leptin acts in the hypothalamus to regulate appetite. If the retroaction works, the regulation of appetite is expected to decrease appetite and food intake, and maintain BMI to a normal level. The knock-outed mice for leptin or congenital leptin deficiency in humans result in uncontrolled satiety and obesity. A bidirectional MR of Steiger filtering in the forward MR could answer this question.

2) In the discussion could the authors say if there are any other examples of already analysed or potentially interesting indirect life course effects ie expand on reference 26? Can they provide other examples of tissue-partitioned effects either already shown or potentially relevant? Can they suggest examples of other issues that this combined approach could address? This could give readers an even better idea of how these new approaches could be used.

*Reviewer #1 (Recommendations for the authors):*

I don't find a lot that needs to be addressed.

I could not find Supplementary notes 1 and 2.

In the discussion could the authors say if there are any other examples of already analysed or potentially interesting indirect life course effects ie expand on reference 26?

Can they provide other examples of tissue-partitioned effects either already shown or potentially relevant?

Any suggestions/examples for other issues that this combined approach could address?

This could give readers an even better idea of how these new approaches could be used.

*Reviewer #2 (Recommendations for the authors):*

This is an interesting study exploring lifecourse effects of adiposity on leptin levels, and using tissue-partitioned MR instruments to study the effects of BMI on leptin levels. While the study question is appealing, the methods are novel and they have been nicely applied in the context of the present study, and the results are interesting, I find that certain parts of the manuscript could be further clarified.

Also, it might not fit directly with the proof of concept aim of the manuscript, but this study could be a nice opportunity to test the opposite association, ie if leptin affects adiposity at different life stages. The relationship between BMI and leptin levels is complex as nicely depicted in lines 248-253 of the discussion. In the introduction, it is stated that leptin acts in the hypothalamus to regulate appetite. If the retroaction works, the regulation of appetite is expected to decrease appetite and food intake, and maintain BMI to a normal level. The knock-outed mice for leptin or congenital leptin deficiency in humans result in uncontrolled satiety and obesity. A bidirectional MR of Steiger filtering in the forward MR could answer this question.

Below are my point-by-point comments:

Abstract:

Methods: When the authors say in the results: "along the causal pathway", do they mean that they adjusted for adult body size in multivariable MR? If yes, please add this to the methods section of the abstract.

The units of change in exposure that confer the betas in the leptin outcome are not exposed in the abstract or in the manuscript (see additional comment on this below). This raises a challenge in directly comparing the two betas (ie in the childhood vs adulthood MR analysis).

In the conclusion, the phrase "as well as raising implications for the gene-environment equivalence assumption in MR". It is not clear how this phrase is justified by the analyses presented in the abstract. It is explained in the discussion of the paper, but I would suggest omitting it from the abstract conclusion.

Some specifications are needed in regards to the exposure and the outcome of GWAS sources.

For the exposure GWAS: In the BMJ paper cited as the source of the SNPs for adult and childhood body size, it is described that adult body size was measured based on the measured body mass index during adulthood (mean age 56.5) while child body mass was self-reported perceived body size at age 10. This is important information that should appear in the methods section of the manuscript. This leads to the question, could the fact that the IVs for measuring adult vs child body mass were based on a different method of estimating body size explain the difference we see in the results of the lifecourse MR? I understand that these IVs successfully predicted adult vs child BMI in 3 cohorts, but the measure of the exposure conferred by the genetic instrument in adults vs children is different. This should be highlighted in the methods, Results section and as a limitation of this study.

In terms of the outcome GWAS:

In line 105, could the authors specify how were leptin levels measured in ALSPAC? Was the method the same as that in DECODE? If the method was different, could this as well induce any bias in the MR estimates?

In terms of comparison between ALSPAC and DECODE, the Icelandic population of DECODE may have a distinct genetic architecture from ALSPAC or other European cohorts. This could introduce bias due to population stratification when using instruments for the exposure that comes from European meta-analyses. Is there another adult proteomic GWAS with available leptin levels which can be used to replicate the results in DECODE?

Line 138: in the multivariable MR analyses, which were the other variables tested as exposures (ie other than the adiposity variants for children or adults?) -this part should be clarified in both methods and Results section. I understand that adulthood and childhood body size were the two exposure variables, based on the DAGs in Figure 1. It would be helpful to put the DAGs in the two boxes entitled "Genetic variants" the GWAS source of the genetic variants and specify: " genetic variants for adulthood adiposity" or "genetic variants for childhood adiposity" depending on the analysis. Also, could the authors explain in the legend the interpretation of the colours of the arrows (ie black vs red?).

Line 144: Could the authors specify if in the tissue partitioned MR analysis the IVs were for adult BMI?

Line 167: in the phrase: "We firstly estimated the total effect of childhood adiposity on circulating leptin using data measured in ALSPAC participants at mean age 9.9 years in the lifecourse", please specify that you are referring to a univariate MR analysis.

In Tables S3 and S5, please provide the p-value of the intercept of the MR-Egger, as well as the Cohrane Q p-values for IVW and MR-Egger, and based on these results, please comment on the presence or absence of evidence of pleiotropy in the IVs used in the MR analyses.

Discussion: It would be nice to further elaborate on the hypothesis explaining the finding in regards to the indirect effect of childhood adiposity on adult leptin levels. In line 239, the authors explain this finding by "individuals in a population remaining overweight for many years in the lifecourse". The effect of early-life adiposity remains after adjustment for adult adiposity. Can the results of the tissue-partitioned MR (in terms of the sole effect of brain BMI SNPs on leptin levels in childhood vs a composite effect of adipose and brain tissue IVs on adult leptin levels) inform us on a possible hypothesis explaining the above phenomenon? Could early-life adiposity be associated with specific hypothalamic responses persisting in adulthood?

In regards to the sensitivity analysis on the effect of tissue-partitioned BMI on visceral vs subcutaneous fat, the authors could provide further explanation on the relationship between leptin secretion and fat distribution providing a rationale for this analysis.

---

## [Author Response]

Essential revisions:1) It might not fit directly with the proof of concept aim of the manuscript, but this study could be a nice opportunity to test the opposite association, ie if leptin affects adiposity at different life stages. The relationship between BMI and leptin levels is complex as nicely depicted in lines 248-253 of the discussion. In the introduction, it is stated that leptin acts in the hypothalamus to regulate appetite. If the retroaction works, the regulation of appetite is expected to decrease appetite and food intake, and maintain BMI to a normal level. The knock-outed mice for leptin or congenital leptin deficiency in humans result in uncontrolled satiety and obesity. A bidirectional MR of Steiger filtering in the forward MR could answer this question.

Many thanks for this suggestion. We have now conducted an additional analysis to evaluate the effect of circulating leptin levels on adulthood BMI using a cis-acting protein quantitative trait loci (pQTL) located in the *LEP* gene as an instrumental variable:

**Author response table 1. sa2table1:** 

Study	Exposure	Outcome	Β	SE	P
deCODE	Circulating leptin	BMI	0.024	0.039	0.541

This provided comparatively weaker effect estimates between circulating leptin levels in adulthood on BMI. pQTL data from the deCODE study was used for this analysis as there were no genome-wide corrected pQTL (i.e. P<5x10-8) in the Fenland study at this locus.

However, as the reviewer mentions this work is outside the scope of this current manuscript. Further research investigating this hypothesis is therefore necessary to robustly evaluate time-varying and tissue-dependent relationships once the relevant data become accessible.

2) In the discussion could the authors say if there are any other examples of already analysed or potentially interesting indirect life course effects ie expand on reference 26? Can they provide other examples of tissue-partitioned effects either already shown or potentially relevant? Can they suggest examples of other issues that this combined approach could address? This could give readers an even better idea of how these new approaches could be used.

We have added references to site-specific cancer endpoints where our lifecourse approach has previously provided evidence of an indirect effect of childhood adiposity (page 15):

“Evidence supporting indirect effects using this approach, as in this study for adulthood leptin, have been found previously for various cardiovascular (30) and site-specific cancer endpoints (31, 32).”

As recommended by Reviewer #2, we have also added some discussion regarding interpretation of findings from both approaches together (page 16):

“Taken together with the evidence of an indirect effect highlighted by our lifecourse MR analysis, these findings suggest that leptin may have long term consequences for appetite suppression. This in turn has an influence on excess body weight which may start in childhood and then can be sustained into later life.”

We have also expanded on further applications of this approach to page 17:

“In particular, applications of a combined approach using both lifecourse and tissue-partitioned MR will likely be most powerful for exposures which have been analysed by GWAS in large samples at distinct timepoints in the lifecourse (e.g. childhood and adulthood) where the functionally important tissue types have been well characterized by previous studies.”

Reviewer #1 (Recommendations for the authors):I don't find a lot that needs to be addressed.I could not find Supplementary notes 1 and 2.

Many thanks for your comments to help refine this manuscript. We have now incorporated Supplementary Notes 1 and 2 into the main article file as requested by the editorial team at *eLife*.

In the discussion could the authors say if there are any other examples of already analysed or potentially interesting indirect life course effects ie expand on reference 26?

We have now added examples of indirect lifecourse effects as suggested to page 15 of the manuscript:

“Evidence supporting indirect effects using this approach, as in this study for adulthood leptin, have been found previously for various cardiovascular (30) and site-specific cancer endpoints (31, 32).”

Can they provide other examples of tissue-partitioned effects either already shown or potentially relevant?

Other examples of tissue-partitioned effects have now been described on page 17:

“For example, we previously found evidence that the brain tissue-derived variants are predominantly responsible for the effect of BMI on both cigarette smoking and lung cancer (8).”

Any suggestions/examples for other issues that this combined approach could address?This could give readers an even better idea of how these new approaches could be used.

We have expanded on further applications of this approach to page 17:

“In particular, applications of a combined approach using both lifecourse and tissue-partitioned MR will likely be most powerful for exposures which have been analysed by GWAS in large samples at distinct timepoints in the lifecourse (e.g. childhood and adulthood) where the functionally important tissue types have been well characterized by previous studies.”

Reviewer #2 (Recommendations for the authors):This is an interesting study exploring lifecourse effects of adiposity on leptin levels, and using tissue-partitioned MR instruments to study the effects of BMI on leptin levels. While the study question is appealing, the methods are novel and they have been nicely applied in the context of the present study, and the results are interesting, I find that certain parts of the manuscript could be further clarified.Also, it might not fit directly with the proof of concept aim of the manuscript, but this study could be a nice opportunity to test the opposite association, ie if leptin affects adiposity at different life stages. The relationship between BMI and leptin levels is complex as nicely depicted in lines 248-253 of the discussion. In the introduction, it is stated that leptin acts in the hypothalamus to regulate appetite. If the retroaction works, the regulation of appetite is expected to decrease appetite and food intake, and maintain BMI to a normal level. The knock-outed mice for leptin or congenital leptin deficiency in humans result in uncontrolled satiety and obesity. A bidirectional MR of Steiger filtering in the forward MR could answer this question.

Many thanks for this suggestion. We have now conducted an additional analysis to evaluate the effect of circulating leptin levels on adulthood BMI using a cis-acting protein quantitative trait loci (pQTL) located in the *LEP* gene as an instrumental variable:

This provided comparatively weaker effect estimates between circulating leptin levels in adulthood on BMI. pQTL data from the deCODE study was used for this analysis as there were no genome-wide corrected pQTL (i.e. P<5x10-8) in the Fenland study at this locus.

However, we believe this work is outside the scope of this current manuscript. Further research investigating this hypothesis is therefore necessary to robustly evaluate time-varying and tissue-dependent relationships once the relevant data become accessible.

Below are my point-by-point comments:Abstract:Methods: When the authors say in the results: "along the causal pathway", do they mean that they adjusted for adult body size in multivariable MR? If yes, please add this to the methods section of the abstract.

We have clarified this point on page 2 as suggested:

“Genetic instruments for childhood and adult adiposity were incorporated into a multivariable MR (MVMR) framework to estimate lifestage specific effects on leptin levels measured during early life (mean age:10 years) in the Avon Longitudinal Study of Parents and Children (ALSPAC) and in adulthood (mean age:55 years) using summary-level data from the deCODE Health study.”

The units of change in exposure that confer the betas in the leptin outcome are not exposed in the abstract or in the manuscript (see additional comment on this below). This raises a challenge in directly comparing the two betas (ie in the childhood vs adulthood MR analysis).

Units of change have now been added to the abstract (1 standard deviation change in circulating leptin levels). The reviewer makes a valid point that using SD change makes comparisons between childhood and adulthood estimates challenging given that the measures of leptin derived from these cohorts will differ in terms of how they vary across populations. We have emphasized this point on page 8:

“Although effect estimates from these summary-level data can be interpreted as a 1-SD change in normalized plasma leptin levels, we note that direct comparisons between estimates derived from childhood measures of leptin in ALSPAC should not be drawn given the various sources of heterogeneity between these measures.”

In the conclusion, the phrase "as well as raising implications for the gene-environment equivalence assumption in MR". It is not clear how this phrase is justified by the analyses presented in the abstract. It is explained in the discussion of the paper, but I would suggest omitting it from the abstract conclusion.

We have omitted this section of the abstract as recommended.

Some specifications are needed in regards to the exposure and the outcome of GWAS sources.For the exposure GWAS: In the BMJ paper cited as the source of the SNPs for adult and childhood body size, it is described that adult body size was measured based on the measured body mass index during adulthood (mean age 56.5) while child body mass was self-reported perceived body size at age 10. This is important information that should appear in the methods section of the manuscript. This leads to the question, could the fact that the IVs for measuring adult vs child body mass were based on a different method of estimating body size explain the difference we see in the results of the lifecourse MR? I understand that these IVs successfully predicted adult vs child BMI in 3 cohorts, but the measure of the exposure conferred by the genetic instrument in adults vs children is different. This should be highlighted in the methods, Results section and as a limitation of this study.

We have added further detail to the childhood and adult body size instruments section as suggested on page 6:

“Briefly, genetic instruments derived in the UK Biobank study based on self-reported body size at age 10 and measured adulthood body mass index (11) have been shown to separate clinically measured childhood and adult BMI in 3 independent cohorts (ALSPAC (11), the Young Finns Study (14) and the Trøndelag Health (HUNT) study (15)).”

We have also added details to the methods section to describe analyses conducted in our original paper which suggested that recall bias related to the childhood measure is unlikely to have influenced the conclusions of our lifecourse MR approach. This point has also been highlighted in our discussion on page 17:

“Moreover, the childhood adiposity instruments are based on recall data, which is why they required validation in 3 independent cohorts as described in our methods section.”

In terms of the outcome GWAS:In line 105, could the authors specify how were leptin levels measured in ALSPAC? Was the method the same as that in DECODE? If the method was different, could this as well induce any bias in the MR estimates?

Further detail to leptin measures analysed in this study are now reported on page 7 and 8. We have also emphasized that the SomaLogic measure of circulating leptin previously provided a high degree of concordance with an alternative measure using the Olink platform, reinforcing its reliability as a measure of this protein given that Olink measures have previously been validated using the same technology as the one used in the ALSPAC study (immunosorbent assay (ELISA)):

“Leptin was measured by an in-house enzyme-linked immunosorbent assay (ELISA) validated against commercial methods. Analyses in ALSPAC were undertaken on a final sample size of 4,155 individuals after removing those without genetic data and withdrawn consent.”

“Genetic estimates on adulthood circulating leptin levels were obtained from a study of 10,708 individuals enrolled in the Fenland study. Full details have been described elsewhere (24). Leptin was measured using the SomaScan v4 assay which applies single-stranded oligonucleotides aptamers with specific binding affinities to protein targets. This assay has also been reported to provide a very highly correlated measure of circulating leptin using the antibody based Olink platform (r=0.95) (25). The Fenland study was approved by the National Health Service (NHS) Health Research Authority Research Ethics Committee (NRES Committee-East of England Cambridge Central, ref. 04/Q018/19). All participants provided written informed consent.”

In terms of comparison between ALSPAC and DECODE, the Icelandic population of DECODE may have a distinct genetic architecture from ALSPAC or other European cohorts. This could introduce bias due to population stratification when using instruments for the exposure that comes from European meta-analyses. Is there another adult proteomic GWAS with available leptin levels which can be used to replicate the results in DECODE?

Many thanks for this suggestion. Since this manuscript was submitted, a protein GWAS study conducted in the Fenland study has made their full summary statistics available (Pietzner et al., Science (2021)). We have there now undertaken an additional analysis using Leptin data derived from the UK based Fenland cohort which we have included as our primary analysis, with deCODE analyses still included as a further validation analysis.

Line 138: in the multivariable MR analyses, which were the other variables tested as exposures (ie other than the adiposity variants for children or adults?) -this part should be clarified in both methods and Results section. I understand that adulthood and childhood body size were the two exposure variables, based on the DAGs in Figure 1. It would be helpful to put the DAGs in the two boxes entitled "Genetic variants" the GWAS source of the genetic variants and specify: " genetic variants for adulthood adiposity" or "genetic variants for childhood adiposity" depending on the analysis. Also, could the authors explain in the legend the interpretation of the colours of the arrows (ie black vs red?).

We have now clarified in the figure legends that multivariable MR analyses requires genetic variants for all exposures to be modelled simultaneously (as well as interpretation regarding the colours of arrows).

Line 144: Could the authors specify if in the tissue partitioned MR analysis the IVs were for adult BMI?

We have now clarified on page 6 that these IVs were based on adult BMI.

Line 167: in the phrase: "We firstly estimated the total effect of childhood adiposity on circulating leptin using data measured in ALSPAC participants at mean age 9.9 years in the lifecourse", please specify that you are referring to a univariate MR analysis.

We have updated this sentence accordingly (page 9):

“We firstly conducted univariable MR to estimate the total effect of childhood adiposity on circulating leptin using data measured in ALSPAC participants at mean age 9.9 years in the lifecourse (Β=1.28 per 1-standard deviation (SD) change in leptin levels per change in body size category, 95% CI=1.10 to 1.46, P=2x10^-41^).”

In Tables S3 and S5, please provide the p-value of the intercept of the MR-Egger, as well as the Cohrane Q p-values for IVW and MR-Egger, and based on these results, please comment on the presence or absence of evidence of pleiotropy in the IVs used in the MR analyses.

These additional statistics have been added to Supplementary Tables and referenced on page 12:

“Cochran’s Q-statistics for these analyses, as well as those derived in Lifecourse MR analyses, can be found in Supplementary File 1l, along with intercept terms for MR-Egger analyses which did not suggest horizontal pleiotropy may be biasing conclusions.”

Discussion: It would be nice to further elaborate on the hypothesis explaining the finding in regards to the indirect effect of childhood adiposity on adult leptin levels. In line 239, the authors explain this finding by "individuals in a population remaining overweight for many years in the lifecourse". The effect of early-life adiposity remains after adjustment for adult adiposity. Can the results of the tissue-partitioned MR (in terms of the sole effect of brain BMI SNPs on leptin levels in childhood vs a composite effect of adipose and brain tissue IVs on adult leptin levels) inform us on a possible hypothesis explaining the above phenomenon? Could early-life adiposity be associated with specific hypothalamic responses persisting in adulthood?

We have added some further discussion regarding this point to page 16:

“Taken together with the evidence of an indirect effect highlighted by our lifecourse MR analysis, these findings suggest that leptin may have long term consequences for appetite suppression. This in turn has an influence on excess body weight which may start in childhood and then can be sustained into later life.”

In regards to the sensitivity analysis on the effect of tissue-partitioned BMI on visceral vs subcutaneous fat, the authors could provide further explanation on the relationship between leptin secretion and fat distribution providing a rationale for this analysis.

Discussion regarding this point can be found on page 8:

“We postulate that these findings highlight a role for appetite regulation and energy expenditure mechanisms as of fundamental importance in the effect of adiposity on leptin levels.”